# GMM-Based Adaptive Extended Kalman Filter Design for Satellite Attitude Estimation under Thruster-Induced Disturbances

**DOI:** 10.3390/s23094212

**Published:** 2023-04-23

**Authors:** Taeho Kim, Natnael S. Zewge, Hyochoong Bang, Hyosang Yoon

**Affiliations:** Department of Aerospace Engineering, Korea Advanced Institute of Science and Technology, 201 Daehak-ro, Yuseong-gu, Daejeon 34141, Republic of Korea; taeho.k@kaist.ac.kr (T.K.);

**Keywords:** satellite attitude estimation, thruster-induced disturbance, blurred star image, Gaussian mixture model, adaptive extended Kalman filter, non-Gaussian noise

## Abstract

Star images from star trackers are usually defocused to capture stars over an exposure time for better centroid measurements. While a satellite is maneuvering, the star point on the screen of the camera is affected by the satellite, which results in the degradation of centroid measurement accuracy. Additionally, this could result in a worse star vector outcome. For geostationary satellites, onboard thrusters are used to maintain or change orbit parameters under orbit disturbances. Since there is misalignment in the thruster and torque is generated by an impulsive shape signal from the torque command, it is difficult to generate target torque; in addition, it also impacts the star image because the impulsive torque creates a sudden change in the angular velocity in the satellite dynamics. This makes the noise of the star image non-Gaussian, which may require introducing a method for dealing with non-Gaussian measurement noise. To meet this goal, in this study, an adaptive extended Kalman filter is implemented to predict measurement vectors with predicted states. The GMM (Gaussian mixture model) is connected in this sequence, giving weighting parameters to each Gaussian density and resulting in the better prediction of measurement vectors. Simulation results show that the GMM-EKF exhibits a better performance than the EKF for attitude estimation, with 30% improvement in performance. Therefore, the GMM-EKF could be a more attractive approach for use with geostationary satellites during station-keeping maneuvers.

## 1. Introduction

This paper introduces a method for adaptively improving the performance of an attitude extended Kalman filter with a star tracker and gyroscope under thruster-induced disturbance onboard a geostationary satellite.

Due to external disturbances such as the influence of the Moon, the Sun and the noncentrality of the Earth’s gravitational field, satellite orbit parameters change with time and, therefore, the orbit ceases to be geostationary [1]. Thrusters onboard geostationary satellites are used to maintain orbit parameters. However, misalignment in the thrusters’ setup produces additional disturbances to the satellite. These problems result in unnecessary fuel consumption during mission mode [2]. Moreover, the torque command from the attitude control law is converted into an impulse shape signal. The pulse width modulator (PWM) impacts the attitude estimation carried out with an extended Kalman filter because star tracker images are directly influenced by the thruster-induced disturbance. In this environment, the noise of a star vector is not white Gaussian noise, but rather is considered to be non-Gaussian noise. Special care should be taken when dealing with non-Gaussian noise if an EKF is the estimator. However, previous research about station-keeping maneuvering for geostationary satellites has not included precise attitude determination under non-Gaussian noise when using star tracker as an attitude sensor [3,4]. Star images are usually generated by defocusing the camera images in order to precisely acquire the centroids of stars. When the satellite is maneuvering, the star image is blurred during the star tracker’s exposure time and the centroid measurements of the star tracker will have degraded accuracy.

There are two categories of solutions when the blurring occurs in the star images. One is to remove the blur directly from the image; the other is to measure the star vector of the blurred image and apply it to the attitude Kalman filter considering a non-Gaussian noise process.

There have already been substantial efforts to remove the blurring effect of star images when under the dynamical conditions of satellites. Most researchers have concentrated on modeling the PSF as accurately as possible and applied it to the deblurring algorithms. The authors of [5] employed a correlation filter to conduct denoising and improve the signal for deblurring star images, considering the angular velocity of the spacecraft to be a constant speed. In addition, the authors of [6] proposed a method to model the PSF of star images and compensate for it under a constant angular velocity and nonfixed angular velocity. Meanwhile, [7] developed a method to simulate multiple blurred star images with uniform and nonuniform blur. Another approach involved restoring blurred star images using maximum likelihood estimation with the aid of microelectromechanical systems (MEMS) gyroscopes [8]. The authors of [9] proposed a motion blur model for the real star tracker that accounts for composite motion beyond uniform and nonuniform motions, and simulated blurred star images under rotations and angular vibrations. Finally, a study by [10] implemented a Kalman filter to estimate the centroids of star images, which improves performance by proposing the covariance prediction equation, adaptive tuning process, and measurement noise matrices depending on the star light magnitude or star existence in the image.

A lot of work has also been done on using the extended Kalman filter (EKF) to estimate the satellite attitude with the star vector measurement as well as recovering blurry images. Ref. [11] represented the algorithm of attitude EKF using a quaternion parameter. Ref. [12] expanded the algorithm of [11] to have the multiplicative error quaternion to avoid a constraint of the quaternion. Since the attitude EKF in [11] assumed a zero-mean Gaussian white-noise process, an adaptive Kalman filter should be used to adaptively tune the parameters in real time if the measurement noise is the non-Gaussian noise process.

The adaptive Kalman filter has been widely studied to tune the process noise matrix and the measurement noise matrix. There are various methods for making the Kalman filter adaptive [13]. A covariance matching algorithm is one of the techniques that is tracking the innovation profile of the filter. It could be divided into the R-adaptation and the Q-adaptation. A multiple model-based adaptive estimation (MMAE) method is also the technique of the adaptive Kalman filter that constructs the Kalman filters with different models and merges the estimates of all filters using the probability that each model is true [14].

However, these techniques have not reflected the non-Gaussian noise process of the measurement. In order to deal with the non-Gaussian noise process, the GMM could be adopted for the prediction of measurement noise with the state of the filter and it provides desirable estimation results [15]. In addition, the GMM has been previously used to compensate for non-Gaussian process noise in the system [16]. Ref. [17] used the GMM to capture the nonlinearities of the light-curve measurement model with the adaptive unscented Kalman filter for the attitude determination. Therefore, the GMM-based extended Kalman filter is applied to predict the non-Gaussian process noise under the blurred star image in this paper.

In this paper, our focus is on increasing the performance of attitude estimation based on star vector measurements with the aid of a gyroscope. The attitude EKF is introduced to carry out the vector measurements via attitude prediction (a priori attitude), utilizing the angular velocity estimate. Since the attitude EKF only considers a measurement model with Gaussian noise, a method that deals with the PSF (point spread function) in star images should be used. A PSF under a small angular velocity would not be a problem for attitude estimation; however, the attitude estimation performance may deteriorate under fast maneuvering or complicated dynamics.

The rest of the paper is organized as follows: Section 2 presents a star image generation method with the PSF, demonstrates thruster modeling, and describes the influence of thruster-induced disturbance on the star image using a simulation example. Section 3 introduces the attitude extended Kalman filter with an error quaternion. Section 4 and Section 5 detail GMM implementation with the extended Kalman filter to predict the non-Gaussian measurement noise. The simulation results of the EKF and GMM-EKF are compared and discussed. These comparisons show that the accuracy of the attitude estimation of the GMM-EKF is around 30% better than that of the EKF.

## 2. Star Image Generation with PSF

### 2.1. Pinhole Camera Model of Star Tracker

The star tracker measurement method is modeled using a pinhole camera model shown in Figure 1. Star points are generated in the focal plane through the optical lens system.

The relationship between the inertial frame and the star tracker frame of a star position coordinate system was obtained from [18], such that
(1)$$b=\frac{1}{\sqrt{{\left(x-x_{0}\right)}^{2}+{\left(y-y_{0}\right)}^{2}+f^{2}}}\left[\begin{array}{c} -\left(x-x_{0}\right)\\ -\left(y-y_{0}\right)\\ f \end{array}\right],$$
(2)$$r=\left[\begin{array}{c} \cos\Lambda_{n}\cos{𝓁}_{n}\\ \sin\Lambda_{n}\cos{𝓁}_{n}\\ \sin{𝓁}_{n} \end{array}\right],$$
where O−xyz is the frame system of the star tracker, $\left(x_{0},y_{0}\right)$ is a point where the boresight axis intersects the focal plane, f is the focal length, b represents the observed star vector in the star tracker frame, r is the reference star vector in the star catalogue that the star tracker is equipped with, and $\left(\Lambda ,𝓁\right)$ is the right ascension and declination of the observed star defined in the celestial sphere. The reference star vector and observed star vector have a relationship with the attitude direction cosine matrix $A\left(q\right)$, which is defined by two coordinate systems written as
(3)$$b=A\left(q\right)r,$$
where $A\left(q\right)$ is the direction cosine matrix for the attitude quaternion q given by
(4)$$q={\left[\begin{array}{cc} \varsigma^{T} & q_{4} \end{array}\right]}^{T},$$
(5)$$\varsigma ={\left[\begin{array}{ccc} q_{1} & q_{2} & q_{3} \end{array}\right]}^{T},$$
where ς corresponds to a vector part of q and $q_{4}$ is a scalar part of q.

### 2.2. Star Tracker Image Generation

Once the star position is determined, the star’s light distribution around the center of the star spot should be calculated. In order to facilitate the determination of the centroids with subpixel accuracy, the optics of the star tracker need to be slightly defocused so that the star light is spread out over several pixels [19]. The most accurate centroiding algorithms rely on fitting a PSF to the measured pixel data [20]. In [21], the star spot PSF is defined by
(6)$${℧}_{\mathrm{PSF}}(x,y)=\frac{1}{2\pi \sigma_{\mathrm{lens}}^{2}}\exp\left(-\frac{{\left(x-x_{0}\right)}^{2}+{\left(y-y_{0}\right)}^{2}}{2\sigma_{\mathrm{lens}}{}^{2}}\right),$$
where $\sigma_{\mathrm{lens}}$ is the Gaussian PSF radius, which is related to the spread scale of the optical lens. In addition, the star’s light distribution for a star spot is defined by
(7)$$E_{\mathrm{spot}}(x,y)=n_{\mathrm{flux}}T_{\exp}{℧}_{\mathrm{PSF}}(x,y),$$
where $n_{\mathrm{flux}}$ is the incident flux of the star on the image plane and $T_{\exp}$ is the exposure time.

In order to obtain the centroid from the star’s light distribution, the center of gravity is evaluated around the star spot as given by [22]
(8)$$\left(x_{c},y_{c}\right)=\left(\frac{\sum_{pi=1}^{}\sum_{pj=1}^{}x_{pi}E_{\mathrm{spot}}(pi,pj)}{\sum_{pi=1}^{}\sum_{pj=1}^{}E_{\mathrm{spot}}(pi,pj)},\;\;\;\frac{\sum_{pi=1}^{}\sum_{pj=1}^{}y_{pi}E_{\mathrm{spot}}(pi,pj)}{\sum_{pi=1}^{}\sum_{pj=1}^{}E_{\mathrm{spot}}(pi,pj)}\right),$$
where $x_{pi}$ and $y_{pi}$ are the $pi^{\mathrm{th}}$ pixel integer coordinates.

### 2.3. Star Tracker Image under Thruster-Induced Disturbance

#### 2.3.1. Thruster Modeling

In general, six thrusters are needed to allow for attitude maneuvers in space; although, some highly sophisticated systems claim to achieve the same space maneuvers with only four thrusters that are strategically located on the satellite body. For various practical reasons six or more thrusters are necessary to complete a reaction control system [23]. Therefore, six thrusters were chosen as the number of thruster units in this paper.

For a single thruster unit, the torque components were derived by considering the setup location, direction of the thruster, and elevation and azimuth angles defined in the coordinate system of the thruster [23]. Figure 2 presents a single thruster’s setup direction with regard to the satellite body axis and sequences of rotation of the thruster’s frame. In this paper, the system of rotation of each thruster unit is the same as the system in reference [23]. First, the $y_{B}$ axis is rotated based on the amount of $\beta_{\mathrm{thr}}$, and then the $z_{B}$ axis is rotated based on the amount of $\alpha_{\mathrm{thr}}$. Hence, the resultant force components are given by
(9)$$\begin{array}{l} F_{\mathrm{thr},x}=F_{\mathrm{lev}}\cos(\alpha_{\mathrm{thr}})\cos(\beta_{\mathrm{thr}}),\;\;\;\\ F_{\mathrm{thr},y}=F_{\mathrm{lev}}\sin(\alpha_{\mathrm{thr}}),\;\;\;\\ F_{\mathrm{thr},z}=F_{\mathrm{lev}}\cos(\alpha_{\mathrm{thr}})\sin(\beta_{\mathrm{thr}}), \end{array}$$
where $F_{\mathrm{thr},x}$, $F_{\mathrm{thr},y}$, and $F_{\mathrm{thr},z}$ are components of the unit thruster force vector $F_{\mathrm{thr}}$, and $F_{\mathrm{lev}}$ represents the thruster level.

The misalignment of the thruster setup is considered in this paper, which leads to
(10)$$\begin{array}{l} F_{\mathrm{thr},x}=F_{\mathrm{lev}}\cos(\alpha_{\mathrm{thr}}+\Delta \alpha_{\mathrm{thr}})\cos(\beta_{\mathrm{thr}}+\Delta \beta_{\mathrm{thr}}),\;\;\;\\ F_{\mathrm{thr},y}=F_{\mathrm{lev}}\sin(\alpha_{\mathrm{thr}}+\Delta \alpha_{\mathrm{thr}}),\;\;\;\\ F_{\mathrm{thr},z}=F_{\mathrm{lev}}\cos(\alpha_{\mathrm{thr}}+\Delta \alpha_{\mathrm{thr}})\sin(\beta_{\mathrm{thr}}+\Delta \beta_{\mathrm{thr}}), \end{array}$$
where $\Delta \alpha_{\mathrm{thr}}$ and $\Delta \beta_{\mathrm{thr}}$ are misalignment angles of the unit thruster setup. By considering the position of the unit thruster $r_{\mathrm{thr}}$, the torque $\tau_{\mathrm{thr}}$ from the unit thruster is given by
(11)$$\begin{array}{l} \tau_{\mathrm{thr}}=r_{\mathrm{thr}}\times F_{\mathrm{thr}}\\ \;\;\;\;\;\;\;\;=\left[\begin{array}{c} r_{\mathrm{thr},y}\sin(\beta_{\mathrm{thr},\mathrm{mis}})\cos(\alpha_{\mathrm{thr},\mathrm{mis}})-r_{\mathrm{thr},z}\sin(\alpha_{\mathrm{thr},\mathrm{mis}})\\ r_{\mathrm{thr},z}\cos(\alpha_{\mathrm{thr},\mathrm{mis}})\cos(\beta_{\mathrm{thr},\mathrm{mis}})-r_{\mathrm{thr},x}\cos(\alpha_{\mathrm{thr},\mathrm{mis}})\sin(\alpha_{\mathrm{thr},\mathrm{mis}})\\ r_{\mathrm{thr},x}\sin(\alpha_{\mathrm{thr},\mathrm{mis}})-r_{\mathrm{thr},y}\cos(\alpha_{\mathrm{thr},\mathrm{mis}})\cos(\beta_{\mathrm{thr},\mathrm{mis}}) \end{array}\right]F_{\mathrm{lev}}\\ \;\;\;\;\;\;\;\;=\left[\begin{array}{c} \Delta x_{\mathrm{th},\mathrm{arm}}\\ \Delta y_{\mathrm{th},\mathrm{arm}}\\ \Delta z_{\mathrm{th},\mathrm{arm}} \end{array}\right]F_{\mathrm{lev}}, \end{array}$$
where $\Delta x_{\mathrm{th},\mathrm{arm}}$, $\Delta y_{\mathrm{th},\mathrm{arm}}$, and $\Delta z_{\mathrm{th},\mathrm{arm}}$ represent the equivalent torque arms of the thruster $\tau_{\mathrm{thr}}$ for the three-axis satellite body frame, and $r_{\mathrm{thr},x}$, $r_{\mathrm{thr},y}$, and $r_{\mathrm{thr},z}$ are the three-axis components of position vector $r_{\mathrm{thr}}$ measured from the center of the mass of the satellite. In addition, $\alpha_{\mathrm{thr},\mathrm{mis}}$ and $\beta_{\mathrm{thr},\mathrm{mis}}$ are rotation angles for considering misalignments given by
(12)$$\begin{array}{l} \alpha_{\mathrm{thr},\mathrm{mis}}=\alpha_{\mathrm{thr}}+\Delta \alpha_{\mathrm{thr}},\\ \beta_{\mathrm{thr},\mathrm{mis}}=\beta_{\mathrm{thr}}+\Delta \beta_{\mathrm{thr}}. \end{array}$$

#### 2.3.2. Thruster Torque Command Generation with Pulse Width Modulation

Reaction controllers can be used in a quasilinear mode by modulating the width of the activated reaction pulse proportionally to the level of the torque command that is input into the controller, which is the often-used pulse width modulation (PWM) principle [23]. Torque from the thruster is generated based on the ratio between the time that the thruster is on and the thruster sampling time. The activating time for each thruster is derived using the thruster set-up model, which is shown in Figure 3. All thruster setups for each thruster location and rotation direction, as well as all equations in this section, were obtained from [23].

The relationship between the three-axis torque command from the control law and the ratio is given by
(13)$$\begin{array}{l} \tau_{\mathrm{cmd},x}=r_{\mathrm{ratio},5}G_{x_{B},5}+r_{\mathrm{ratio},3}G_{x_{B},3}-r_{\mathrm{ratio},4}G_{x_{B},4}-r_{\mathrm{ratio},6}G_{x_{B},6}\\ \tau_{\mathrm{cmd},z}=r_{\mathrm{ratio},5}G_{z_{B},5}+r_{\mathrm{ratio},6}G_{z_{B},6}-r_{\mathrm{ratio},3}G_{z_{B},3}-r_{\mathrm{ratio},4}G_{z_{B},4}\\ \tau_{\mathrm{cmd},y}=r_{\mathrm{ratio},2}G_{y_{B},2}-r_{\mathrm{ratio},1}G_{y_{B},1}, \end{array}$$
where $r_{\mathrm{ratio},i}$ represents the $i^{\mathrm{th}}\left(i=1,2,\;\;\ldots \;\;,6\right)$ ratio and $G_{x_{B},i}$, $G_{y_{B},i}$, and $G_{z_{B},i}$ are $i^{\mathrm{th}}$ torque constants written as
(14)$$\begin{array}{l} G_{x_{B},i}=F_{\mathrm{lev}}\Delta x_{\mathrm{th},\mathrm{arm},i}\\ G_{y_{B},i}=F_{\mathrm{lev}}\Delta y_{\mathrm{th},\mathrm{arm},i}\\ G_{z_{B},i}=F_{\mathrm{lev}}\Delta z_{\mathrm{th},\mathrm{arm},i} \end{array}$$
where, $\Delta x_{\mathrm{th},\mathrm{arm},i}$, $\Delta y_{\mathrm{th},\mathrm{arm},i}$, and $\Delta z_{\mathrm{th},\mathrm{arm},i}$ are the $i^{\mathrm{th}}$ torque arms of the three-axis satellite body frame. Using the thruster setup in Figure 3, the torque constants are defined as
(15)$$\begin{array}{l} G_{x_{B},3}=G_{x_{B},4}=G_{x_{B},5}=G_{x_{B},6}=F_{\mathrm{lev}}\Delta x_{\mathrm{th},\mathrm{arm},\mathrm{ex}}=G_{x_{B},\mathrm{ex}}\\ G_{z_{B},3}=G_{z_{B},4}=G_{z_{B},5}=G_{z_{B},6}=F_{\mathrm{lev}}\Delta z_{\mathrm{th},\mathrm{arm},\mathrm{ex}}=G_{z_{B},\mathrm{ex}}\\ G_{y_{B},1}=G_{y_{B},2}=F_{\mathrm{lev}}\Delta y_{\mathrm{th},\mathrm{arm},\mathrm{ex}}=G_{y_{B},\mathrm{ex}} \end{array}$$
where $\Delta x_{\mathrm{th},\mathrm{arm},\mathrm{ex}}$, $\Delta y_{\mathrm{th},\mathrm{arm},\mathrm{ex}}$, and $\Delta z_{\mathrm{th},\mathrm{arm},\mathrm{ex}}$ represent the torque arms of the three-axis satellite body frame, and the torque constants for each axis are represented as $G_{x_{B},\mathrm{ex}}$, $G_{y_{B},\mathrm{ex}}$, and $G_{z_{B},\mathrm{ex}}$, as shown in Figure 3. Then, Equation (13) can be rewritten as
(16)$$\begin{array}{l} \tau_{\mathrm{cmd},x}=\left(r_{\mathrm{ratio},5}+r_{\mathrm{ratio},3}-r_{\mathrm{ratio},4}-r_{\mathrm{ratio},6}\right)G_{x_{B},\mathrm{ex}}\\ \tau_{\mathrm{cmd},z}=\left(r_{\mathrm{ratio},5}+r_{\mathrm{ratio},6}-r_{\mathrm{ratio},3}-r_{\mathrm{ratio},4}\right)G_{z_{B},\mathrm{ex}}\\ \tau_{\mathrm{cmd},y}=\left(r_{\mathrm{ratio},2}-r_{\mathrm{ratio},1}\right)G_{y_{B},\mathrm{ex}}. \end{array}$$

After normalization of the command torques, Equation (16) becomes
(17)$$\begin{array}{l} {\hat{\tau }}_{\mathrm{cmd},x}=\frac{\tau_{\mathrm{cmd},x}}{G_{x_{B},\mathrm{ex}}}=r_{\mathrm{ratio},5}+r_{\mathrm{ratio},3}-r_{\mathrm{ratio},4}-r_{\mathrm{ratio},6}\\ {\hat{\tau }}_{\mathrm{cmd},z}=\frac{\tau_{\mathrm{cmd},z}}{G_{z_{B},\mathrm{ex}}}=r_{\mathrm{ratio},5}+r_{\mathrm{ratio},6}-r_{\mathrm{ratio},3}-r_{\mathrm{ratio},4}\\ {\hat{\tau }}_{\mathrm{cmd},y}=\frac{\tau_{\mathrm{cmd},y}}{G_{y_{B},\mathrm{ex}}}=r_{\mathrm{ratio},2}-r_{\mathrm{ratio},1}, \end{array}$$
where ${\hat{\tau }}_{\mathrm{cmd},x}$, ${\hat{\tau }}_{\mathrm{cmd},y}$, and ${\hat{\tau }}_{\mathrm{cmd},z}$ are the normalized torques for each axis. The first and second normalized torques from Equation (17) can be rewritten in a matrix form as
(18)$$\left[\begin{array}{c} {\hat{\tau }}_{\mathrm{cmd},x}\\ {\hat{\tau }}_{\mathrm{cmd},z} \end{array}\right]=\left[\begin{array}{cccc} 1 & -1 & 1 & -1\\ -1 & -1 & 1 & 1 \end{array}\right]\left[\begin{array}{c} r_{\mathrm{ratio},3}\\ r_{\mathrm{ratio},4}\\ r_{\mathrm{ratio},5}\\ r_{\mathrm{ratio},6} \end{array}\right].$$

By evaluating a pseudoinverse, the above equation can be rewritten as
(19)$$\left[\begin{array}{c} r_{\mathrm{ratio},3}\\ r_{\mathrm{ratio},4}\\ r_{\mathrm{ratio},5}\\ r_{\mathrm{ratio},6} \end{array}\right]=\frac{1}{4}\left[\begin{array}{cc} 1 & -1\\ -1 & -1\\ 1 & 1\\ -1 & 1 \end{array}\right]\left[\begin{array}{c} {\hat{\tau }}_{\mathrm{cmd},x}\\ {\hat{\tau }}_{\mathrm{cmd},z} \end{array}\right].$$

The thruster on-time for each thruster could be found to be negative based on the three-axis torque command, which is not physically possible. In order to solve this problem, the thruster unit which has a negative on-time is turned off and replaced with the one which has a positive on-time, enabling it to provide the same torque. The operational logic is shown in Algorithm 1 [23].

**Algorithm 1:** Thruster on-time setting algorithm

$$\begin{array}{l} \mathrm{Input}:\;{\hat{\tau }}_{\mathrm{cmd},x},\;{\hat{\tau }}_{\mathrm{cmd},y},\;{\hat{\tau }}_{\mathrm{cmd},z}\\ \mathrm{Compute the ratio of thruster 3, 4, 5, 6 using (19).}\\ \;rr_{\mathrm{ratio},6}=r_{\mathrm{ratio},6}-r_{\mathrm{ratio},3}\;;\;\;\;rr_{\mathrm{ratio},3}=0\;;\\ \;\mathrm{if}\;\;\left(rr_{\mathrm{ratio},6}<0\right)\;\;rr_{\mathrm{ratio},3}=r_{\mathrm{ratio},3}-r_{\mathrm{ratio},4};\;\;\;rr_{\mathrm{ratio},6}=0;\;\;\;\mathrm{end}\\ \;rr_{\mathrm{ratio},4}=r_{\mathrm{ratio},4}-r_{\mathrm{ratio},5}\;;\;\;\;rr_{\mathrm{ratio},5}=0\;;\\ \;\mathrm{if}\;\;\left(rr_{\mathrm{ratio},4}<0\right)\;\;rr_{\mathrm{ratio},5}=r_{\mathrm{ratio},5}-r_{\mathrm{ratio},4};\;\;\;rr_{\mathrm{ratio},4}=0;\;\;\;\mathrm{end}\\ \;\mathrm{if}\;\;\left({\hat{\tau }}_{\mathrm{cmd},y}>0\right)\;\;rr_{\mathrm{ratio},2}={\hat{\tau }}_{\mathrm{cmd},y};\;\;\;rr_{\mathrm{ratio},1}=0\;\;;\;\mathrm{end}\\ \;\mathrm{if}\;\;\left({\hat{\tau }}_{\mathrm{cmd},y}<0\right)\;\;rr_{\mathrm{ratio},1}=\mathrm{Abs}\left({\hat{\tau }}_{\mathrm{cmd},y}\right);\;\;\;rr_{\mathrm{ratio},2}=0\;\;;\;\mathrm{end}\\ \mathrm{Output}:\;r_{\mathrm{ratio},1},\;r_{\mathrm{ratio},2},\;r_{\mathrm{ratio},3},\;r_{\mathrm{ratio},4},\;r_{\mathrm{ratio},5},\;r_{\mathrm{ratio},6}\\ \left[\begin{array}{cccccc} r_{\mathrm{ratio},1} & r_{\mathrm{ratio},2} & r_{\mathrm{ratio},3} & r_{\mathrm{ratio},4} & r_{\mathrm{ratio},5} & r_{\mathrm{ratio},6} \end{array}\right]=\\ \;\;\;\;\;\;\;\;\;\;\;\;\;\;\;\;\;\;\;\;\;\;\;\;\;\;\;\;\;\;\;\;\;\;\;\;\;\;\;\;\;\;\;\;\;\;\;\;\;\;\;\;\;\left[\begin{array}{cccccc} rr_{\mathrm{ratio},1} & rr_{\mathrm{ratio},2} & rr_{\mathrm{ratio},3} & rr_{\mathrm{ratio},4} & rr_{\mathrm{ratio},5} & rr_{\mathrm{ratio},6} \end{array}\right] \end{array}$$



In Algorithm 1, $\mathrm{Abs}\left(\;\cdot \;\right)$ represents an absolute of the value.

#### 2.3.3. Star Image Implementation under Thruster-Induced Disturbance

In this section, a star image under thruster-induced disturbance is simulated to model the smearing effect and is compared with a situation where the satellite is stationary. For the simulation, the thruster setup is the same as that shown in Figure 3. Thruster specifications are presented in Table 1 and star tracker specifications are provided in Table 2. The star tracker boresight axis is aligned with the $z_{B}$ axis of the satellite body frame.

The satellite is assumed to be initially stationary, and the goal of this simulation is to maintain its attitude as zero. To control the satellite attitude, a quaternion feedback PD control law is applied in a form obtained from [24] as given by
(20)$$u=-\left[\omega \times \right]J\omega -K_{D}\omega -K_{P}q_{e},$$
where ω is an angular velocity vector of the body frame relative to the reference frame and $\left[\omega \times \right]$ is a skew symmetric matrix of ω. J represents the moment of inertia of the satellite and $q_{e}$ is the error of the current and target quaternion defined by
(21)$$q_{e}=q⊗q_{\mathrm{target}}{}^{-1},$$
where $K_{D}$ and $K_{P}$ are control gains which are written as
(22)$$\begin{array}{l} K_{D}=d_{\mathrm{gain}}J,\;\;\\ K_{P}=k_{\mathrm{gain}}J, \end{array}$$
where $d_{\mathrm{gain}}$ and $k_{\mathrm{gain}}$ are designed to be in the form of
(23)$$\begin{array}{l} d_{\mathrm{gain}}=2\xi_{\mathrm{dam}}\omega_{n},\\ k_{\mathrm{gain}}=2\omega_{n}^{2}, \end{array}$$
where $\xi_{\mathrm{dam}}$ is a damping ratio and $\omega_{n}$ is a natural frequency given by
(24)$$\omega_{n}=\frac{8}{t_{s}\xi_{dam}},$$
where $t_{s}$ is a settling time. The rigid satellite model and controller gain information is presented in Table 3.

Three stars were captured by the star tracker at the initial attitude in the simulation. The magnitudes of each star were 4.24, 3.03, and 4.52. The simulation time was 100 s, and these three stars were continuously captured in the camera of the star tracker. During the simulation, the true quaternion and true angular velocity were used for attitude control without sensor data since the goal was to examine the smearing effect with the angular velocity induced by thruster disturbance.

Euler angle error, angular velocity, and thruster torque are described in Figure 4, Figure 5 and Figure 6, respectively.

Figure 4 shows that the Euler angle has a steady state error, which is related to the thruster’s sampling time and the closed-loop bandwidth. The angular velocity has an oscillatory profile in Figure 5 because the output of each thruster is generated in an impulse form, with the sampling time of the thruster as shown in Figure 6a.

The smearing effect due to the thruster torque is described in Figure 7 and Figure 8. At the initial time, since the angular velocity is 0.01°/s for each axis, the star image is not blurred as much by the angular motion as it is in the thruster-operating case. In Figure 8, each star spot is more influenced by the angular motion than in Figure 7.

In order to reduce the smearing effect, an attitude extended Kalman filter will be introduced in the next section that predicts measurement noise based on the PSF information.

## 3. Extended Kalman Filter for Attitude Determination

In this section, an extended Kalman filter for spacecraft attitude determination is introduced. Observations from multiple stars are used to carry out measurements with the EKF and angular velocity measurements from the gyroscope are used for the propagation of states and covariances of the EKF. The quaternion parameter is chosen to represent the spacecraft attitude because it is free from singularities for all attitudes. In addition, the quaternion features the lowest dimensional attitude parameterization compared to all other alternatives. However, the quaternion has a normalization constraint which may be violated during the update sequence of the standard EKF. Instead of using the quaternion as a state, a multiplicative error quaternion-based extend Kalman filter is selected because the four-component quaternion can effectively be replaced by a three-component error vector [25].

### 3.1. Multiplicative Quaternion Formulation

A multiplicative quaternion-based extended Kalman filter, made by Lefferts et al. [12], has been used to implement an attitude determination filter. This section briefly introduces the derivation and configuration of this filter by referring to [12] and [25]. The derivation of the multiplicative extended Kalman filter begins with a quaternion kinematics model described as
(25)$$\dot{q}=\frac{1}{2}\Omega \left(\omega \right)q,$$
where
(26)$$\Omega \left(\omega \right)=\left[\begin{array}{cc} -\left[\omega \times \right] & \omega \\ -\omega^{T} & 0 \end{array}\right],$$
and q is a quaternion where $q={\left[\begin{array}{cc} \varsigma^{T} & q_{4} \end{array}\right]}^{T}$. Because of the normalization constraint of the quaternion, error quaternion kinematics is adopted for the extended Kalman filter.

First, an error quaternion is defined as
(27)$$\delta q=q⊗{\hat{q}}^{-1},$$
where $\delta q={\left[\begin{array}{cc} \delta \varsigma^{T} & \delta q_{4} \end{array}\right]}^{T}$ and ⊗ is an operator for quaternion multiplication. Error quaternion kinematics is written as
(28)$$\delta \dot{q}=-\left[\begin{array}{c} \left[\hat{\omega }\times \right]\delta \varsigma \\ 0 \end{array}\right]+\frac{1}{2}\left[\begin{array}{c} \delta \omega \\ 0 \end{array}\right]⊗\delta q.$$

Following first-order approximation, it is given by
(29)$$\begin{array}{l} \delta \dot{\varsigma }=-\left[\hat{\omega }\times \right]\delta \varsigma +\frac{1}{2}\delta \omega ,\\ \delta {\dot{q}}_{4}=0. \end{array}$$

A rate-integrating gyro is a commonly used sensor for measuring angular velocity and its observation model is defined as
(30)$$\begin{array}{l} \omega =\tilde{\omega }-\beta -\eta_{v},\\ \dot{\beta }=\eta_{u}. \end{array}$$
where $\tilde{\omega }$ is a gyroscope measurement, β is a bias vector, and $\eta_{v}$ and $\eta_{u}$ are zero-mean Gaussian white-noise processes with covariances given by $\sigma_{v}^{2}I_{3\times 3}$ and $\sigma_{u}^{2}I_{3\times 3}$, respectively [25]. The estimated angular velocity and the time derivative of the bias vector are as follows:(31)$$\begin{array}{l} \hat{\omega }=\tilde{\omega }-\hat{\beta },\\ \dot{\hat{\beta }}=0. \end{array}$$
with Equations (30) and (31), and $\delta \omega =\omega -\hat{\omega }$, Equation (29) yields
(32)$$\delta \dot{\varsigma }=-\left[\hat{\omega }\times \right]\delta \varsigma -\frac{1}{2}\left(\Delta \beta +\eta_{v}\right),$$
where $\Delta \beta =\beta -\hat{\beta }$. The small angle approximation, δς=δα/2, gives
(33)$$\delta \alpha =-\left[\hat{\omega }\times \right]\delta \alpha -\left(\Delta \beta +\eta_{v}\right),$$
where δα is the components of the roll, pitch, and yaw angles for any rotational sequence. Using Equations (30) and (32), the extended Kalman filter error model is described as
(34)$$\Delta \dot{\tilde{x}}(t)=F(\hat{x}(t),t)\Delta \tilde{x}(t)+G(t)w(t),$$
where $\Delta \tilde{x}(t)={\left[\begin{array}{cc} \delta \alpha^{T}(t) & \delta \beta^{T}(t) \end{array}\right]}^{T}$, $w(t)={\left[\begin{array}{cc} \eta_{v}^{T}(t) & \eta_{u}^{T}(t) \end{array}\right]}^{T}$, and $F(\hat{x}(t),t)$. Additionally, G(t) and Q(t) are given by
(35)$$F(\hat{x}(t),t)=\left[\begin{array}{cc} -\left[\hat{\omega }(t)\times \right] & -I_{3\times 3}\\ 0_{3\times 3} & 0_{3\times 3} \end{array}\right],$$
(36)$$G(t)=\left[\begin{array}{cc} -I_{3\times 3} & 0_{3\times 3}\\ 0_{3\times 3} & 0_{3\times 3} \end{array}\right],$$
(37)$$Q(t)=\left[\begin{array}{cc} \sigma_{v}^{2}I_{3\times 3} & 0_{3\times 3}\\ 0_{3\times 3} & \sigma_{u}^{2}I_{3\times 3} \end{array}\right].$$

The discrete time–star vector observation matrix is defined using the current quaternion and reference star vector in the star catalogue and is given by
(38)$${\tilde{y}}_{k}=\left[\begin{array}{c} A(q)r_{1}\\ A(q)r_{2}\\ \vdots \\ A(q)r_{n} \end{array}\right]+\left[\begin{array}{c} \nu_{1}\\ \nu_{2}\\ \vdots \\ \nu_{n} \end{array}\right],$$
where k is a sample index, ${\tilde{y}}_{k}$ is a measurement matrix, A(q) is a direction cosine matrix of the current attitude, $r_{n}$ is the reference vector in the star catalogue of the $n^{\mathrm{th}}$ observed vector, and $\nu_{n}$ is a zero-mean Gaussian white-noise process of the $n^{\mathrm{th}}$ observed vector with a covariance of $\sigma_{n}^{2}I_{3\times 3}$, which leads to a measurement covariance matrix such that
(39)$$R_{k}=\mathrm{diag}\left[\begin{array}{cccc} \sigma_{1}^{2}I_{3\times 3} & \sigma_{2}^{2}I_{3\times 3} & \ldots & \sigma_{n}^{2}I_{3\times 3} \end{array}\right].$$

The sensitivity matrix for the star vector measurements has the form
(40)$$H_{k}\left({\hat{x}}_{k}^{-}\right)=\left[\begin{array}{cc} \left[A({\hat{q}}^{-})r_{1}\times \right] & 0_{3\times 3}\\ \left[A({\hat{q}}^{-})r_{2}\times \right] & 0_{3\times 3}\\ \vdots & \vdots \\ \left[A({\hat{q}}^{-})r_{n}\times \right] & 0_{3\times 3} \end{array}\right].$$

The estimate output is given by
(41)$$h_{k}\left({\hat{x}}_{k}^{-}\right)=\left[\begin{array}{c} A({\hat{q}}^{-})r_{1}\\ A({\hat{q}}^{-})r_{2}\\ \vdots \\ A({\hat{q}}^{-})r_{n} \end{array}\right].$$

Then, the error-state update is written as
(42)$$\Delta {\hat{\tilde{x}}}_{k}^{+}=K_{k}\left[{\tilde{y}}_{k}-h_{k}\left({\hat{x}}_{k}^{-}\right)\right],$$
where $\Delta {\hat{\tilde{x}}}_{k}^{+}={\left[\begin{array}{cc} \delta {\hat{\alpha }}_{k}^{+}{}^{T} & \delta {\hat{\beta }}_{k}^{+}{}^{T} \end{array}\right]}^{T}$ and $K_{k}$ is given by
(43)$$K_{k}=P_{k}^{-}H_{k}^{T}\left({\hat{x}}_{k}^{-}\right){\left[H_{k}\left({\hat{x}}_{k}^{-}\right)P_{k}^{-}H_{k}^{T}\left({\hat{x}}_{k}^{-}\right)+R_{k}\right]}^{-1}.$$

Using Equation (42), the bias and quaternion updates are given by
(44)$${\hat{\beta }}_{k}^{+}={\hat{\beta }}_{k}^{-}+\Delta {\hat{\beta }}_{k}^{+},$$
(45)$${\hat{q}}_{k}^{+}={\hat{q}}_{k}^{-}+\frac{1}{2}\Xi \left({\hat{q}}_{k}^{-}\right)\delta {\hat{\alpha }}_{k}^{+},$$
where
(46)$$\Xi \left(q\right)=\left[\begin{array}{c} q_{4}I_{3\times 3}+\left[\varsigma \times \right]\\ -\varsigma^{T} \end{array}\right].$$

Renormalization of the quaternion update should be applied to the result of Equation (45).

The propagation of the state and covariance is outlined below. The readers are referred to [25] for detailed derivations. The propagated quaternion is given by
(47)$${\hat{q}}_{k+1}^{-}=\bar{\Omega }\left({\hat{\omega }}_{k}^{+}\right){\hat{q}}_{k}^{+},$$
with
(48)$$\bar{\Omega }\left({\hat{\omega }}_{k}^{+}\right)=\left[\begin{array}{cc} \cos\left(\frac{1}{2}\left\|{\hat{\omega }}_{k}^{+}\right\|\Delta t\right)I_{3\times 3}-\left[{\hat{\psi }}_{k}^{+}\times \right] & {\hat{\psi }}_{k}^{+}\\ -{\hat{\psi }}_{k}^{+}{}^{T} & \cos\left(\frac{1}{2}\left\|{\hat{\omega }}_{k}^{+}\right\|\Delta t\right) \end{array}\right],$$
where
(49)$${\hat{\psi }}_{k}^{+}=\frac{\sin\left(\frac{1}{2}\left\|{\hat{\omega }}_{k}^{+}\right\|\Delta t\right){\hat{\omega }}_{k}^{+}}{\left\|{\hat{\omega }}_{k}^{+}\right\|},$$

The bias propagation and postupdate angular velocity are given as
(50)$$\begin{array}{l} {\hat{\beta }}_{k}^{-}={\hat{\beta }}_{k}^{+}\\ {\hat{\omega }}_{k}^{+}={\tilde{\omega }}_{k}-{\hat{\beta }}_{k}^{+}, \end{array}$$

The propagation of the covariance is given by
(51)$$P_{k+1}^{-}=\Phi_{k}P_{k}^{+}\Phi_{k}^{T}+G_{k}Q_{k}G_{k}^{T},$$
with
(52)$$\Phi_{k}=\left[\begin{array}{cc} \Phi_{k,11} & \Phi_{k,12}\\ \Phi_{k,21} & \Phi_{k,22} \end{array}\right],$$
where
(53)$$\begin{array}{l} \Phi_{k,11}=I_{3\times 3}-\left[{\hat{\omega }}_{k}^{+}\times \right]\frac{\sin\left(\left\|{\hat{\omega }}_{k}^{+}\right\|\Delta t\right)}{\left\|{\hat{\omega }}_{k}^{+}\right\|}+{\left[{\hat{\omega }}_{k}^{+}\times \right]}^{2}\frac{\left\{1-\cos\left(\left\|{\hat{\omega }}_{k}^{+}\right\|\Delta t\right)\right\}}{{\left\|{\hat{\omega }}_{k}^{+}\right\|}^{2}}\\ \Phi_{k,12}=\left[{\hat{\omega }}_{k}^{+}\times \right]\frac{\left\{1-\cos\left(\left\|{\hat{\omega }}_{k}^{+}\right\|\Delta t\right)\right\}}{{\left\|{\hat{\omega }}_{k}^{+}\right\|}^{2}}-{\left[{\hat{\omega }}_{k}^{+}\times \right]}^{2}\frac{\left\{\left\|{\hat{\omega }}_{k}^{+}\right\|\Delta t-\sin\left(\left\|{\hat{\omega }}_{k}^{+}\right\|\Delta t\right)\right\}}{{\left\|{\hat{\omega }}_{k}^{+}\right\|}^{3}}-I_{3\times 3}\Delta t\\ \Phi_{k,21}=0_{3\times 3}\\ \Phi_{k,22}=I_{3\times 3}, \end{array}$$

The process noise covariance is given by
(54)$$Q_{k}=\left[\begin{array}{cc} \left(\sigma_{v}^{2}\Delta t+\frac{1}{3}\sigma_{u}^{2}\Delta t^{3}\right)I_{3\times 3} & -\left(\frac{1}{2}\sigma_{u}^{2}\Delta t^{2}\right)I_{3\times 3}\\ -\left(\frac{1}{2}\sigma_{u}^{2}\Delta t^{2}\right)I_{3\times 3} & \left(\sigma_{u}^{2}\Delta t\right)I_{3\times 3} \end{array}\right].$$

## 4. Non-Gaussian Measurement Noise Modeling

The multiplicative extended Kalman filter introduced in Section 3 assumes that the noise of the observed star vector is a zero-mean Gaussian white-noise process. Due to thruster-induced disturbance, the star images are blurred, which means this assumption is untrue and leads to the necessity of modeling the measurement error as non-Gaussian noise.

One of the methods for modeling non-Gaussian noise is the GMM. Based on the Wiener approximation theorem, any non-Gaussian noise distribution can be expressed as, or approximated sufficiently well by, a finite sum of known Gaussian distributions [15].

**Lemma** **1** [7]**.** *Any density f(g) associated with an m dimensional vector g can be approximated as closely as desired by a density of the form *
(55)$$f_{C}(g)=\sum_{\zeta =1}^{l}a_{\zeta }N\left(\mu_{\zeta },B_{\zeta }\right).$$ *For some integer* l *and positive scalars* $a_{\zeta }$ *with* $\sum_{\zeta =1}^{l}a_{\zeta }=1$*, where *
$N\left(\cdot \right)$ *is a Gaussian density with mean value* $\mu_{\zeta }$ *and covariance matrix* $B_{\zeta }$*:*
(56)$$N\left(\mu_{\zeta },B_{\zeta }\right)=\frac{1}{{\left(2\pi \right)}^{m}}\det{\left(B_{\zeta }\right)}^{-0.5}\exp\left(-0.5{\left\|g-\mu_{i}\right\|}_{B_{\zeta }^{-1}}^{2}\right),$$*where* ζ *is a Gaussian distribution index*, det(⋅) *is the matrix determinant and* $\left\|\;\cdot \;\right\|$ *is the inner product in the Euclidean space* $R^{m}$.

If the number of Gaussian densities increases, the density function $f_{C}(g)$ may converge and the covariance of each density will approach a zero matrix [15].

GMM-based predicted star vector measurements are constructed by adding $\mu_{\zeta }$ to the predicted star vector measurement (yellow circle), and these are marked with four (l=4) green circles in Figure 9.

## 5. GMM-Based Adaptive Extended Kalman Filter

In the previous section, a Gaussian point generation method for non-Gaussian observed vectors was introduced. Using Equation (55), the non-Gaussian observed vectors can be approximated as a weighted Gaussian mixture. For the configuration of an EKF using the GMM, it is important to find out the optimal Gaussian approximation for the mixture. For this purpose, the linear adaptive Kalman filter algorithm proposed by Plataniotis et al. [15] was referred to for implementing the adaptive attitude estimation extended Kalman filter. For each propagated vector from the observed vectors, the extended Kalman filter is implemented in parallel. Then, based on the interim results from these dedicated Kalman filters, we can obtain a Bayesian a posteriori approximation of the Gaussian mixture required in the filtering process [15]. Finally, the optimal Gaussian approximation for the mixture is obtained by an adaptive algorithm with a weighting evaluation. The equations for the GMM-EKF are introduced below.

In the multiplicative quaternion-based EKF described in Section 3.1, after propagating the states and covariance matrix with the postupdate states and covariance matrix, the innovation and Kalman gain are the next parameters to be obtained. Given the non-Gaussian measurement noise, we have to evaluate the parameters for each Gaussian point and adaptively assemble these parameters before computing the innovation covariance and Kalman gain.

The estimate measurement output is given by
(57)$$h_{k,\mathrm{sum}}=\sum_{\zeta =1}^{N_{\zeta }}\gamma_{k,\zeta }\;h_{k,\zeta },$$
where $h_{k,\mathrm{sum}}$ is a weighted sum of each estimate measurement output with propagated quaternions, $h_{k,\zeta }$ is the $\zeta^{\mathrm{th}}$ GMM-based predicted measurement, and $\gamma_{k,\zeta }$ is a weight for the $\zeta^{\mathrm{th}}$ Gaussian density. In addition, $h_{k,\zeta }$ is defined as
(58)$$h_{k,\zeta }=\left[\begin{array}{c} A({\hat{q}}_{k}^{-})r_{1}\\ A({\hat{q}}_{k}^{-})r_{2}\\ \vdots \\ A({\hat{q}}_{k}^{-})r_{n} \end{array}\right]+\mu_{\zeta }.$$

Here, each group of elements, $h_{k,\zeta }\left(3\kappa -2:3\kappa \right),\;\;\;\kappa =1,\;\;2,\;\;\ldots \;\;n$, should be renormalized because of the constraint that a unit star vector has.

The innovation covariance is given by
(59)$$P_{z}{}_{k}^{-}=\sum_{\zeta =1}^{N_{\zeta }}\left(P_{z}{}_{k,\zeta }^{-}+\left(h_{k,\mathrm{sum}}-h_{k,\zeta }\right)\times {\left(h_{k,\mathrm{sum}}-h_{k,\zeta }\right)}^{T}\right)\;\;\gamma_{k,\zeta },$$
with
(60)$$P_{z}{}_{k,\zeta }^{-}=H_{k,\zeta }P_{k}^{-}H_{k,\zeta }{}^{T}+R_{k,\zeta },$$
where
(61)$$H_{k,\zeta }=\left[\begin{array}{cc} \left[h_{k,\zeta }\left(1:3\right)\times \right] & 0_{3\times 3}\\ \left[h_{k,\zeta }\left(4:6\right)\times \right] & 0_{3\times 3}\\ \vdots & \vdots \\ \left[h_{k,\zeta }\left(3n-2:3n\right)\times \right] & 0_{3\times 3} \end{array}\right].$$
and $R_{k,\zeta }$ is a measurement covariance matrix for the $\zeta^{\mathrm{th}}$ propagated quaternion. Moreover, $\gamma_{k,\zeta }$ is given by
(62)$$\begin{array}{l} \gamma_{k,\zeta }\\ =\frac{\left({\left(2\pi \right)}^{-m}\det{\left(P_{z}{}_{k,\zeta }^{-}\right)}^{-1}\exp\left(-0.5\left({\left({\tilde{y}}_{k}-h_{k,\zeta }\right)}^{T}P_{z}{{}_{k,\zeta }^{-}}^{-1}\left({\tilde{y}}_{k}-h_{k,\zeta }\right)\right)\right)\right)a_{\zeta }}{c_{k}}, \end{array}$$
where $a_{\zeta }$ represents the initial weighting coefficients defined in Equation (55) and $c_{k}$ is a normalization factor given by
(63)$$\begin{array}{l} c_{k}=\sum_{\zeta =1}^{N_{\zeta }}\left({\left(2\pi \right)}^{-m}\det{\left(P_{z}{}_{k,\zeta }^{-}\right)}^{-1}\times \right.\\ \;\;\;\;\;\;\;\;\;\;\;\;\;\;\;\;\;\;\left.\times \exp\left(-0.5\left({\left({\tilde{y}}_{k}-h_{k,\zeta }\right)}^{T}P_{z}{{}_{k,\zeta }^{-}}^{-1}\left({\tilde{y}}_{k}-h_{k,\zeta }\right)\right)\right)\right)a_{\zeta }. \end{array}$$

Then, the Kalman gain is obtained by
(64)$$K_{k,\mathrm{sum}}=P_{k}^{-}H_{k,\mathrm{sum}}^{T}{\left(P_{z}{}_{k}^{-}\right)}^{-1},$$
with
(65)$$H_{k,\mathrm{sum}}=\left[\begin{array}{cc} \left[h_{k,\mathrm{sum}}\left(1:3\right)\times \right] & 0_{3\times 3}\\ \left[h_{k,\mathrm{sum}}\left(4:6\right)\times \right] & 0_{3\times 3}\\ \vdots & \vdots \\ \left[h_{k,\mathrm{sum}}\left(3n-2:3n\right)\times \right] & 0_{3\times 3} \end{array}\right],$$
where $h_{k,\mathrm{sum}}\left(3n-2:3n\right)$ denotes a $\left[3\times 1\right]$ matrix from $h_{k,\mathrm{sum}}$, of which the elements are evaluated using the $n^{\mathrm{th}}$ reference vector. Then, the state update begins with
(66)$$\Delta {\hat{\tilde{x}}}_{k,\mathrm{sum}}^{+}=K_{k,\mathrm{sum}}\left[{\tilde{y}}_{k}-h_{k,\mathrm{sum}}\right],$$
where
(67)$$\Delta {\hat{\tilde{x}}}_{k,\mathrm{sum}}^{+}={\left[\begin{array}{cc} \delta {\hat{\alpha }}_{k,\mathrm{sum}}^{+}{}^{T} & \delta {\hat{\beta }}_{k,\mathrm{sum}}^{+}{}^{T} \end{array}\right]}^{T}.$$

The final processes of the states update are as follows:(68)$$\begin{array}{c} {\hat{\beta }}_{k}^{+}={\hat{\beta }}_{k}^{-}+\Delta {\hat{\beta }}_{k,\mathrm{sum}}^{+},\\ {\hat{q}}_{k}^{+}={\hat{q}}_{k}^{-}+\frac{1}{2}\Xi \left({\hat{q}}_{k}^{-}\right)\delta {\hat{\alpha }}_{k,\mathrm{sum}}^{+},\\ P_{k}^{+}=\left[I-K_{k,\mathrm{sum}}H_{k}\right]P_{k}^{-}. \end{array}$$

The logic of the GMM-EKF is presented in Algorithm 2.

**Algorithm 2:** GMM-EKF algorithm

$$\begin{array}{l} \mathrm{Input}:\;{\hat{q}}_{k-1}^{+},\;{\hat{\beta }}_{k-1}^{+},\;P_{k-1}^{+},\;{\hat{\omega }}_{k-1}^{+},\;a_{\zeta },\;{\tilde{y}}_{k},\;\tilde{\omega },\;\mu_{\zeta },\;Q_{k-1},\;R_{k-1},\;G_{k-1}\\ Propagation:\\ \;{\hat{q}}_{k}^{-}=\bar{\Omega }\left({\hat{\omega }}_{k-1}^{+}\right){\hat{q}}_{k-1}^{+}\\ \;{\hat{\beta }}_{k}^{-}={\hat{\beta }}_{k-1}^{+}\\ \;P_{k}^{-}=\Phi_{k-1}P_{k-1}^{+}\Phi_{k-1}^{T}+G_{k-1}Q_{k-1}G_{k-1}^{T}\\ \mathrm{Gain:}\\ \;\mathrm{Computation}\;\mathrm{and}\;\mathrm{renormalization}\;\mathrm{of}\;h_{k,\zeta }\;\mathrm{as}\;\mathrm{in}\;\mathrm{(58)}\\ \;\mathrm{Computation}\;\mathrm{of}\;H_{k,\zeta }\;\mathrm{as}\;\mathrm{in}\;\mathrm{(61)}\\ \;\mathrm{Computation}\;\mathrm{of}\;P_{z}{}_{k,\zeta }^{-}\;\mathrm{as}\;\mathrm{in}\;\mathrm{(60)}\\ \;\mathrm{Computation}\;\mathrm{of}\;c_{k},\;\gamma_{k,\zeta }\;\mathrm{as}\;\mathrm{in}\;\mathrm{(62)}\;\mathrm{and}\;\mathrm{(63)}\\ \;\mathrm{Computation}\;\mathrm{of}\;h_{k,\mathrm{sum}}\;\mathrm{as}\;\mathrm{in}\;\mathrm{(57)}\\ \;\mathrm{Computation}\;\mathrm{of}\;P_{z}{}_{k}^{-}\;\mathrm{as}\;\mathrm{in}\;\mathrm{(59)}\\ \;\mathrm{Computation}\;\mathrm{of}\;H_{k,\mathrm{sum}}\;\mathrm{as}\;\mathrm{in}\;\mathrm{(65)}\\ \;K_{k,\mathrm{sum}}=P_{k}^{-}H_{k,\mathrm{sum}}^{T}{\left(P_{z}{}_{k}^{-}\right)}^{-1}\\ \mathrm{Update:}\\ \;\Delta {\hat{\tilde{x}}}_{k,\mathrm{sum}}^{+}=K_{k,\mathrm{sum}}\left[{\tilde{y}}_{k}-h_{k,\mathrm{sum}}\right]\\ \;{\hat{\beta }}_{k}^{+}={\hat{\beta }}_{k}^{-}+\Delta {\hat{\beta }}_{k,\mathrm{sum}}^{+}\\ \;{\hat{\omega }}_{k}^{+}={\tilde{\omega }}_{k}-{\hat{\beta }}_{k}^{+}\\ \;{\hat{q}}_{k}^{+}={\hat{q}}_{k}^{-}+\frac{1}{2}\Xi \left({\hat{q}}_{k}^{-}\right)\delta {\hat{\alpha }}_{k,\mathrm{sum}}^{+}\\ \;\mathrm{Computation}\;\mathrm{of}\;H_{k}\;\mathrm{as}\;\mathrm{in}\;\mathrm{(40)}\\ \;P_{k}^{+}=\left[I-K_{k,\mathrm{sum}}H_{k}\right]P_{k}^{-}\\ \mathrm{Output}:\;{\hat{q}}_{k}^{+},\;{\hat{\beta }}_{k}^{+},\;{\hat{\omega }}_{k}^{+},\;P_{k}^{+} \end{array}$$



## 6. Simulation Study

In this section, the GMM-based EKF is implemented using the profile of the quaternion, angular velocity, gyroscope, and star tracker measurements generated for the simulation in Section 2.3.1. The performance of the GMM-EKF and EKF is compared with the same profiles. The gyroscope and filter conditions are shown in Table 4, Table 5 and Table 6.

### 6.1. GMM-EKF Simulation Results

The proposed algorithm was simulated under thruster-induced disturbance. Since the measurement noise is non-Gaussian, a weighted sum of Gaussian densities was introduced to predict the measurement. In this simulation, four Gaussian densities were chosen and added to the predicted measurement with the a priori quaternion. Since quaternion prediction is obtained using the angular velocity estimate, reducing bias from the angular velocity is also important. For each Gaussian density with mean values selected, the pixel errors of the centroid were considered to widely cover the measurement noise. The process noise matrix $Q_{k}$ from Equation (54) was chosen as $10Q_{k}$ due to its better performance. The accuracy of the GMM-EKF estimation error was evaluated using the root mean square error (RMSE), as shown in Equation (69).
(69)$$\mathrm{RMSE}=\sqrt{\frac{1}{n_{\mathrm{total}}}\sum_{k=1}^{n_{\mathrm{total}}}{\left({\hat{x}}_{k}-x_{k}\right)}^{2}}$$
where $n_{\mathrm{total}}$ represents the total number of estimates, ${\hat{x}}_{k}$ represents the estimated state, and $x_{k}$ is the true state.

In Figure 10a, the Euler angle estimate error from the GMM-EKF is shown, in which the RMSE is [3.43, 3.89, 4.86] arcsec. This RMSE corresponds to an almost 0.20-pixel error and is improved compared to the raw pixel error. In Figure 10b, the bias estimate error from the GMM-EKF is shown, in which the RMSE is $\left[6.96,8.03,8.21\right]\times 10^{-4}{}^{\circ}/s$.

A Monte Carlo simulation was conducted to verify the convergence of the GMM-EKF. One hundred sets of initial conditions were simulated, with the initial Euler estimation errors ranging from −0.5 to 0.5° and initial bias estimation errors ranging from $-4.2\times 10^{-3}{}^{\circ}/s$ to $4.2\times 10^{-3}{}^{\circ}/s$. Figure 11 shows the attitude and bias estimation errors for all 100 cases, and it can be observed that all cases converged successfully.

### 6.2. Comparison between GMM-EKF and EKF Performance

The GMM-EKF and EKF are compared in this section. Since the EKF only considers the measurement noise as Gaussian, the predicted measurement is only made with an a priori quaternion. It cannot cover non-Gaussian measurement noise. After tuning the EKF parameters, the process noise $Q_{k}$ was chosen as $10Q_{k}$, and the measurement noise matrix $R_{k}$ was set as $5R_{k}$ due to their better performance. $R_{k}$ was changed in order to make the estimation error stay within the three-sigma boundary. In Figure 12, the RMSE of the Euler angle estimate of the EKF is [5.48, 5.71, 6.09] arcsec, which is equivalent to an almost 0.30-pixel error. Figure 13 shows the attitude estimation error profiles from the GMM-EKF and EKF. Therefore, the GMM-EKF performs better than the EKF under thruster-induced disturbance.

## 7. Conclusions

Because of misalignments in the thrusters of geostationary satellites and the impulse characteristic of their torque, the performance of vector measurements taken with a star tracker decreases as the PSF of the star image is blurred. Thruster modeling and PSF characteristics were investigated and implemented to examine the influence of the thruster-induced disturbance on star images. The blurring effect on images could be seen from our simulation, and thus, this study implemented an attitude extended Kalman filter. In order to handle the non-Gaussian noise of the star vector measurements, the GMM was introduced and implemented with an attitude extended Kalman filter to predict the measurement noise with the aid of a priori knowledge of the attitude quaternion. Four Gaussian densities were chosen by considering the pixel noise from the thruster-induced disturbance and weights for each mixture and were automatically updated depending on the innovation covariance between the star vector measurements and the predicted measurement from the GMM. Simulation results indicate that the GMM-EKF produces a better performance than the EKF in regard to attitude estimation. Therefore, the GMM-EKF could be considered as a potential approach in geostationary satellite missions for attitude estimation under thruster firing.

## Figures and Tables

**Figure 1 sensors-23-04212-f001:**
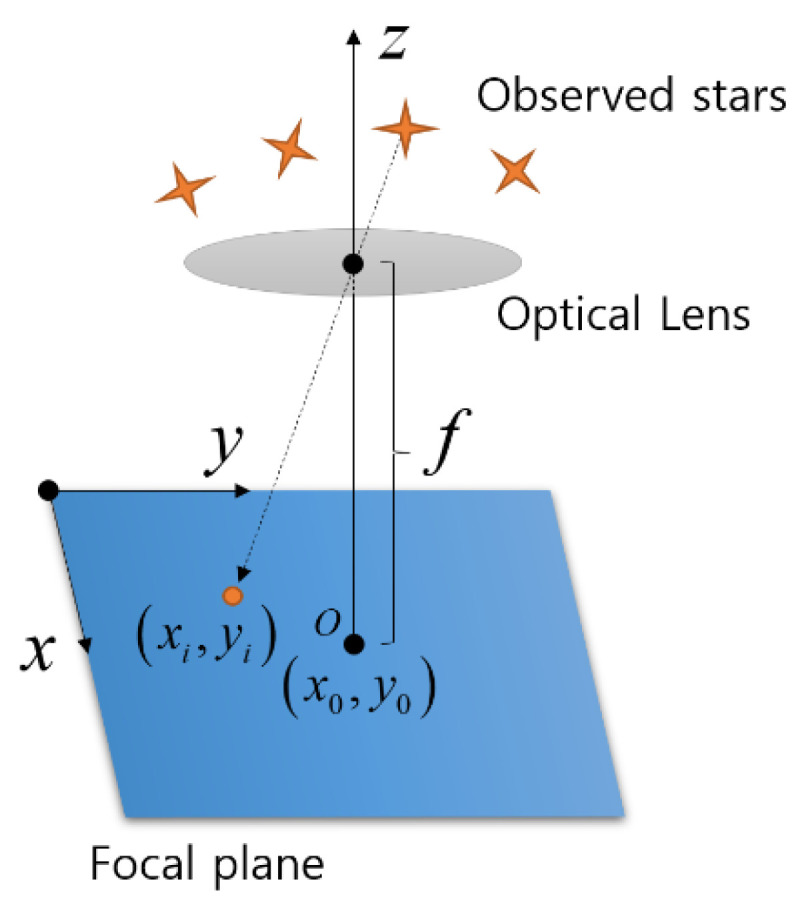
A pinhole camera model of star tracker.

**Figure 2 sensors-23-04212-f002:**
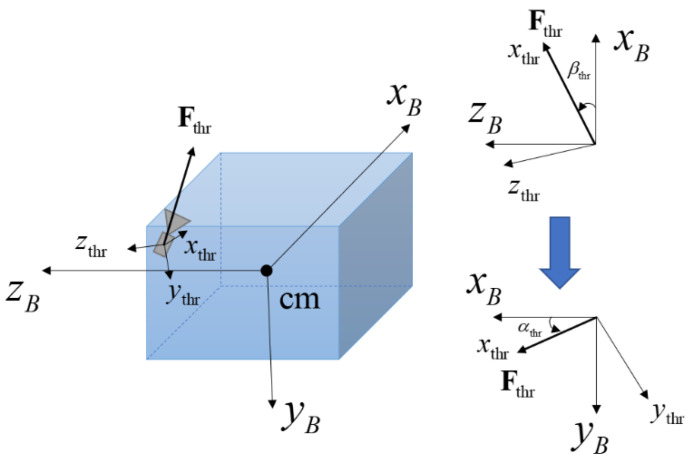
Single thruster’s setup direction regarding body axis.

**Figure 3 sensors-23-04212-f003:**
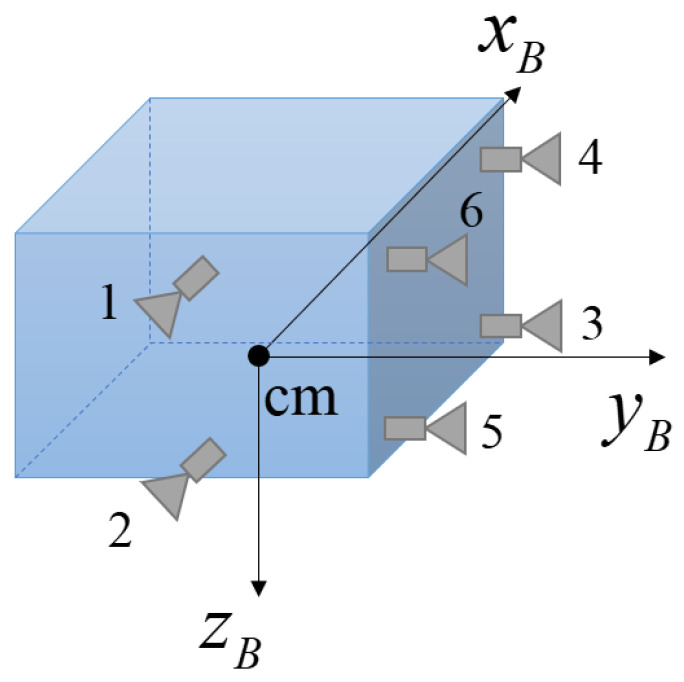
Example thruster setup for a satellite (featuring six thruster units).

**Figure 4 sensors-23-04212-f004:**
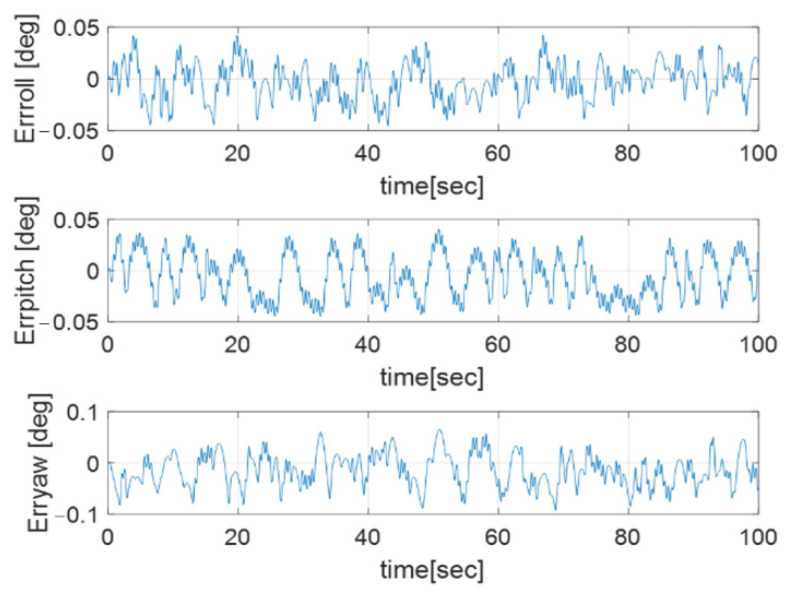
Euler angle error profile.

**Figure 5 sensors-23-04212-f005:**
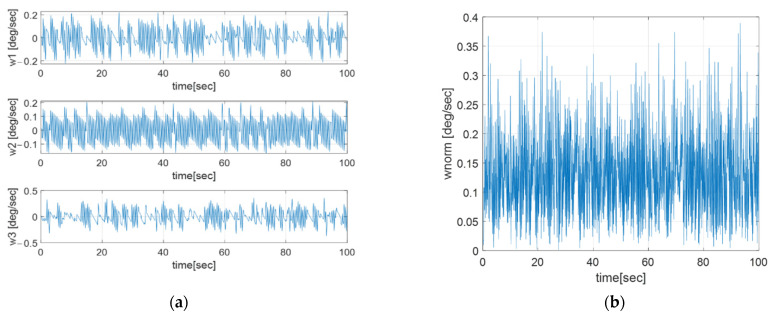
(**a**) Three-axis angular velocity profile; (**b**) Angular velocity norm profile.

**Figure 6 sensors-23-04212-f006:**
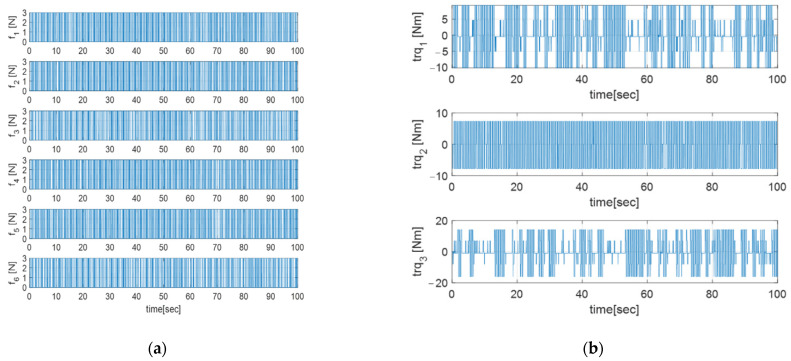
(**a**) Thruster force profile of each thruster unit; (**b**) Three-axis torque profile.

**Figure 7 sensors-23-04212-f007:**
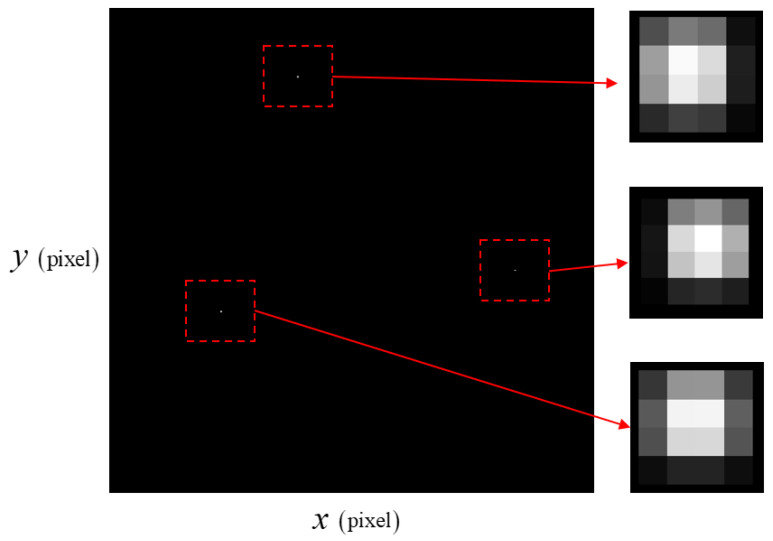
Star image under $\left|\omega \right|=0.0173{}^{\circ}/s$.

**Figure 8 sensors-23-04212-f008:**
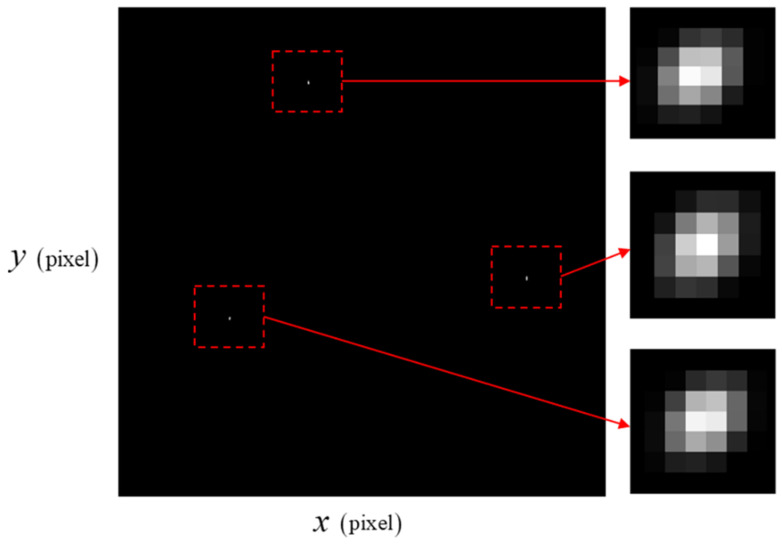
Star image under $\left|\omega \right|=0.3{}^{\circ}/s$.

**Figure 9 sensors-23-04212-f009:**
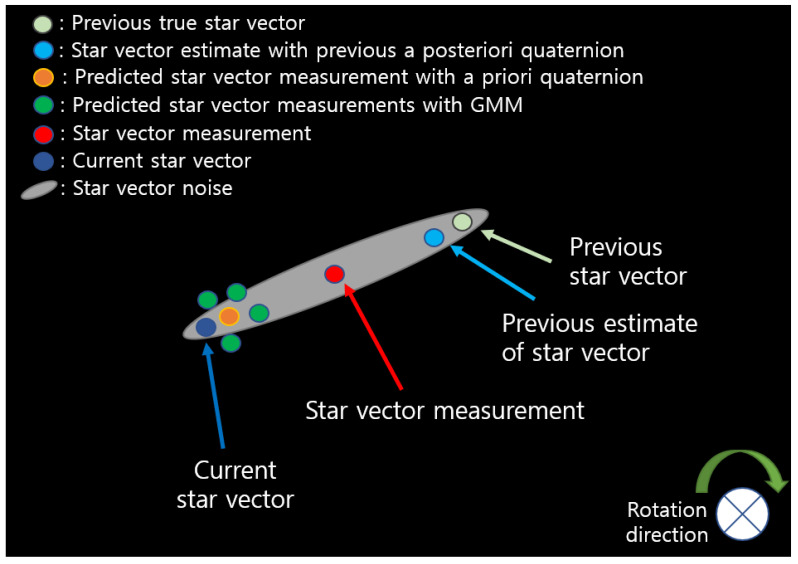
GMM distribution of non-Gaussian noise of star vector measurement.

**Figure 10 sensors-23-04212-f010:**
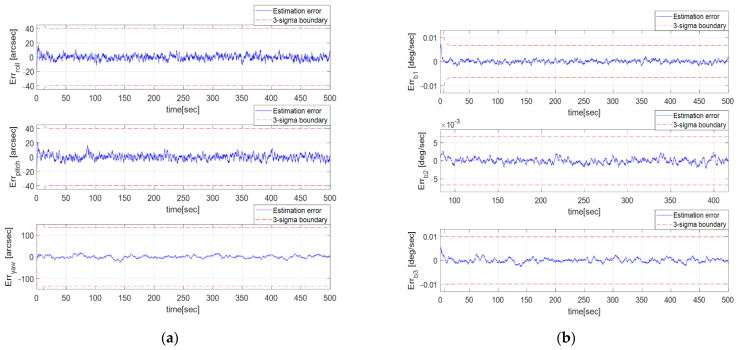
(**a**) Euler estimate error profile of GMM-EKF; (**b**) Bias estimate error profile of GMM-EKF.

**Figure 11 sensors-23-04212-f011:**
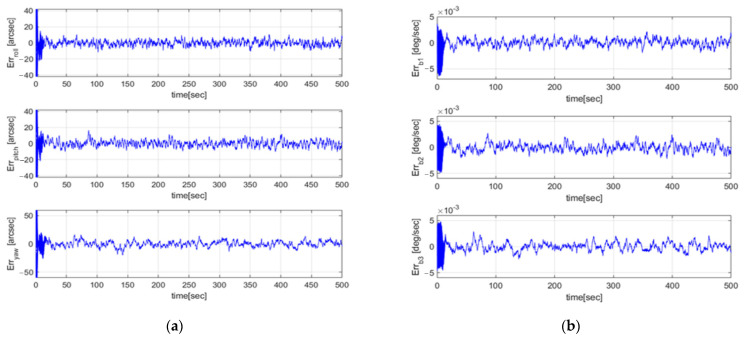
(**a**) Euler estimate error of GMM-EKF (Monte Carlo simulation); (**b**) Bias estimate error profile of GMM-EKF (Monte Carlo simulation).

**Figure 12 sensors-23-04212-f012:**
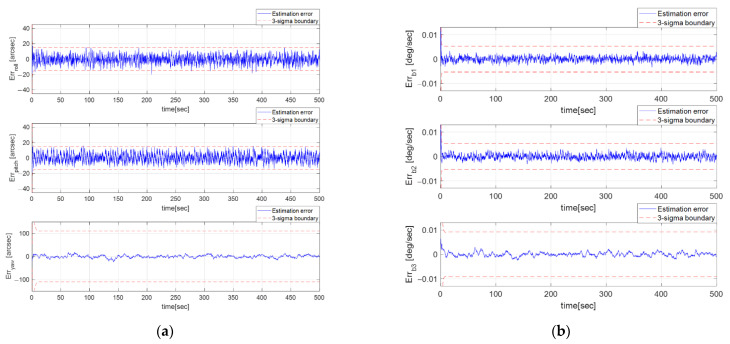
(**a**) Euler estimate error profile of EKF; (**b**) Bias estimate error profile of EKF.

**Figure 13 sensors-23-04212-f013:**
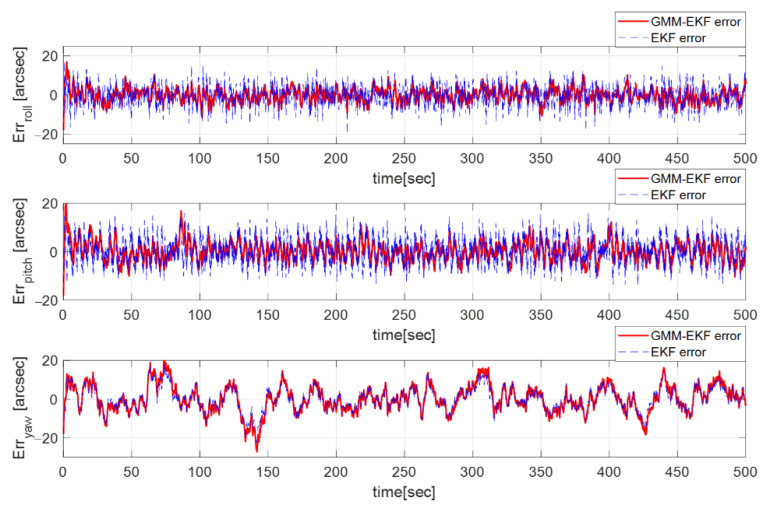
Euler estimate error with GMM-EKF (solid line) and EKF (dashed line).

**Table 1 sensors-23-04212-t001:** Thruster specifications for simulation.

Thruster Index (i)	$\mathrm{Position}\left(r_{\mathrm{thr},x},r_{\mathrm{thr},y},r_{\mathrm{thr},z}\right)$	Elevation $\mathrm{Angle}\;(\alpha_{\mathrm{thr}})$	Azimuth $\mathrm{Angle}\;(\beta_{\mathrm{thr}})$	$\mathrm{Misalignment}\;(\alpha_{\mathrm{thr}}$ $,\beta_{\mathrm{thr}})$	Thruster Level $\left(F_{\mathrm{lev}}\right)$	Sampling Rate
#1	(−1, 0, −0.5) m	30°	90°	3°, 3°	3 N	4 Hz
#2	(−1, 0, 0.5) m	30°	90°			
#3	(1, 1, 0.5) m	0°	30°			
#4	(1, 1, −0.5) m	0°	30°			
#5	(−1, 1, 0.5) m	0°	30°			
#6	(−1, 1, −0.5) m	0°	30°			

**Table 2 sensors-23-04212-t002:** Star tracker specifications for simulation.

Pixel Array Size	Focal Length	Pixel Size	Field of View	Exposure Time	Magnitude Threshold	Radius of Gaussian PSF
1024 × 1024	76.08 mm	13 μm	10° × 10°	100 ms	5	3.8 pixel

**Table 3 sensors-23-04212-t003:** Satellite model and parameters of quaternion feedback PD control law.

Moment of Inertia $\left(I_{xx},I_{yy},I_{zz}\right)$	Damping Ratio	Settling Time	Initial Angular Velocity	Initial Attitude	Target Attitude
[500, 500, 500]$kgm^{2}$	1	2.5 s	[0.01,0.01,0.01]°/s	[0,0,0]°	[0,0,0]°

**Table 4 sensors-23-04212-t004:** Gyroscope specifications for simulation.

Angle Random Walk	Rate Random Walk	Initial Bias	Update Frequency
0.001°/h	0.05${}^{\circ}/h^{3/2}$	[0.0004, −0.0003, 0.0001]°/s	100 Hz

**Table 5 sensors-23-04212-t005:** Filter conditions for simulation.

Measurement Noise Matrix	Update Frequency	$\mathrm{Gaussian}\;\mathrm{Density}\;\mathrm{Mean}\;\mathrm{Value}\;(\mu_{\zeta })$	
$\left[\begin{array}{ccc} 3.5^{2} & 0 & 0\\ 0 & 3.5^{2} & 0\\ 0 & 0 & 3.5^{2} \end{array}\right]\mathrm{arc}\sec^{2}$	100 Hz	−8 arcsec	(For three axes)
		−4 arcsec	
		4 arcsec	
		8 arcsec	

**Table 6 sensors-23-04212-t006:** Filter initial states.

Euler Angle	Bias	Covariance
0.005 ×[1,1,1]°	0.0042×[1,1,1]°/s	$\left[\begin{array}{cccccc} {\left(0.005^{\circ }\right)}^{2} & 0 & 0 & 0 & 0 & 0\\ 0 & {\left(0.005^{\circ }\right)}^{2} & 0 & 0 & 0 & 0\\ 0 & 0 & {\left(0.005^{\circ }\right)}^{2} & 0 & 0 & 0\\ 0 & 0 & 0 & {\left(0.05^{\circ }/\sec\right)}^{2} & 0 & 0\\ 0 & 0 & 0 & 0 & {\left(0.05^{\circ }/\sec\right)}^{2} & 0\\ 0 & 0 & 0 & 0 & 0 & {\left(0.05^{\circ }/\sec\right)}^{2} \end{array}\right]$

## Data Availability

Not applicable.

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
