# Peer review of "GMM-Based Adaptive Extended Kalman Filter Design for Satellite Attitude Estimation under Thruster-Induced Disturbances"

_sensors, 2023, doi:10.3390/s23094212_

Round 1

Reviewer 1 Report

This paper proposes using a GMM-EKF approach to improve the accuracy of star vector measurements during satellite maneuvers. The approach provides better prediction of measurement vectors and exhibits a 30% improvement in performance compared to the traditional EKF approach. 

The method and conclusions are clearly presented, therefore the recommendation is to publish in its present form. 

Author Response

Thank you for the comments. It was helpful to revise the manuscript. We asked the English editing service to Sensor. Most of the grammatical issues have been solved. 

Reviewer 2 Report

In this manuscript, an attitude extended Kalman filter is used to predict the measurement vectors with predicted states. Using a prior knowledge of the attitude equation, the authors implemented an attitude extended Kalman filter to predict the measurement noise to deal with the non-Gaussian noise of the star vector measurements. Simulation results are presented to validate the performance of the proposed algorithm. However, there are some issues to be addressed.

1. It would be better to modify syntax and grammar errors.

2. There are insufficient references in this manuscript, and there are too few research results in the past five years.

3. The Introduction section has less discussion of the latest research results. Therefore, I suggest that the authors supplement it in the coming manuscript.

4. It would be better to present the proposed algorithms by flowchart or pseudocode in the table.

5. On page 19, RMSE is an indicator used to characterize the error, and it is necessary to briefly introduce this indicator.

6. In Figures 10-11, the legend is missing. And authors should use generated rather than intercepted plots.

7. A single 100 second simulation is not enough to reflect the performance of the proposed method. It is better to conduct more experiments for performance comparison to achieve statistical effect. Besides, the algorithm proposed by the authors should be compared with the current state-of-the-art research results in the field to show its advantages.

Author Response

Point 1: It would be better to modify syntax and grammar errors.

Response 1: All syntax and grammar errors have been revised. We asked a revision service to Sensor for English editing.

Point 2: There are insufficient references in this manuscript, and there are too few research results in the past five years.

Response 2: We have supplemented references for introduction part to explain previous research results.

Point 3: The Introduction section has less discussion of the latest research results. Therefore, I suggest that the authors supplement it in the coming manuscript.

Response 3: We have supplemented additional discussion of the lastest research results within the last 5 years.

Point 4: It would be better to present the proposed algorithms by flowchart or pseudocode in the table.

Response 4: We have presented a pseudocode of the proposed algorithms in the table.

Point 5: On page 19, RMSE is an indicator used to characterize the error, and it is necessary to briefly introduce this indicator.

Response 5: The breif introduction of RMSE has been added in the new manuscript.

Point 6: In Figures 10-11, the legend is missing. And authors should use generated rather than intercepted plots.

Response 6: The legend has been added to Figure 10-11. Please explain the meaning of generated and intercepted plots more clearly. We will revise the graph in the next stage.

Point 7: A single 100 second simulation is not enough to reflect the performance of the proposed method. It is better to conduct more experiments for performance comparison to achieve statistical effect. Besides, the algorithm proposed by the authors should be compared with the current state-of-the-art research results in the field to show its advantages.

Response 7: For the simulation of the proposed method, the simulation time has been extended to 500 second. We can extend the simulation time more than 500 second if needed. In addition, we did Monte Carlo simulation with one hundred initial conditions and checked all cases converged. We have studied this subject concentrating on solving the problem which is resulted from using the extended Kalman filter. So, we used Gaussian mixture model (GMM) as a tool for dealing with non-Gaussian noise. Finally, we checked that the GMM-EKF results in better prediction of measurements with non-Gaussian noise and attitude estimation performance compared to the EKF.

Reviewer 3 Report

1. In figure 10 what do the red curves mean?  Is this uncertainty in the estimates deduced from the EKF GMM covariance matrix?  Is this one-sigma or 3-sigma or something else? 

2. In figure 10 please explain why the red curves for bias start at zero at time 0.  How can you achieve zero bias error?  Why does the bias error grow rapidly with time, whereas it was zero at time zero?  This is not consistent.

3. In figure 10 why is the uncertainty so much larger than the actual errors?

4. In figure 11 did you tune the process noise for the EKF to be optimal?  How did you accomplish this tuning?

5. There are many grammatical errors throughout the entire paper.  Very often the word “the” is missing, or the sentence does not make sense.  Please ask someone who is more fluent in English to help you improve the grammar.

Author Response

Point 1: In figure 10 what do the red curves mean?  Is this uncertainty in the estimates deduced from the EKF GMM covariance matrix?  Is this one-sigma or 3-sigma or something else? 

Response 1: We have added a legend in this graph. The red curve in Figure 10 in the previous manuscript was three-sigma boundary for attitude and bias error. This uncertainty in the estimates are deduced from the GMM-EKF covariance matrix.

Point 2: In figure 10 please explain why the red curves for bias start at zero at time 0.  How can you achieve zero bias error?  Why does the bias error grow rapidly with time, whereas it was zero at time zero?  This is not consistent.

Response 2: There was an error in the previous simulation code and it has been fixed. Also, the initial covariance matrix elements for bias was set to too small value. That’s the reason why the red curves for bias seemed to be started at zero at time 0. The intial covariance matrix has been changed for the new simulation. The rapid growth of the bias error has been solved in the revised simulation.

Point 3: In figure 10 why is the uncertainty so much larger than the actual errors?

Response 3: The initial covariance for the filter was set to too small value. Once it has been changed, this problem has been solved.

Point 4: In figure 11 did you tune the process noise for the EKF to be optimal?  How did you accomplish this tuning?

Response 4: We have tuned the process noise matrix and measurement noise matrix so that estimate error is within three-sigma boundary and we can get lower RMSE for each state. We multipled the constant which is ranged from 1~1000 to the theoriotical process noise matrix (Equation (54)) and the constant which is ranged from 1~20 to the measurement noise matrix.

Point 5: There are many grammatical errors throughout the entire paper.  Very often the word “the” is missing, or the sentence does not make sense.  Please ask someone who is more fluent in English to help you improve the grammar.

Response 5: We asked the English editing service to Sensor. Most grammatical issues have been solved.
